# Interplay between Shelf Life and Printability of Silica-Filled Suspensions

**DOI:** 10.3390/polym15214334

**Published:** 2023-11-06

**Authors:** Xavier M. Torres, John R. Stockdale, Santosh Adhikari, Shelbie A. Legett, Adam Pacheco, Jesus A. Guajardo, Andrea Labouriau

**Affiliations:** Los Alamos National Laboratory, Los Alamos, NM 87545, USA; jstockdale@lanl.gov (J.R.S.); san_adh@lanl.gov (S.A.);

**Keywords:** silica, siloxane suspensions, rheology, additive manufacturing, direct ink writing, 3D printability

## Abstract

Although fumed silica/siloxane suspensions are commonly employed in additive manufacturing technology, the interplay between shelf life, storage conditions, and printability has yet to be explored. In this work, direct ink writing (DIW) was used to print unique three-dimensional structures that required suspensions to retain shape and form while being printed onto a substrate. Suspensions containing varying concentrations of hydrophobic and hydrophilic silica were formulated and evaluated over a time span of thirty days. Storage conditions included low (8%) and high (50%) relative humidity and temperatures ranging from 4 °C to 25 °C. The shelf life of the suspensions was examined by comparing the print quality of pristine and aged samples via rheology, optical microscopy, and mechanical testing. Results showed a significant decrease in printability over time for suspensions containing hydrophilic fumed silica, whereas the printability of suspensions containing hydrophobic fumed silica remained largely unchanged after storage. The findings in this work established the following recommendations for extending the shelf life and printability of suspensions commonly used in DIW technology: (1) higher fumed silica concentrations, (2) low humidity and low temperature storage environments, and (3) the use of hydrophobic fumed silica instead of hydrophilic fumed silica.

## 1. Introduction

The rheological behavior of nanoparticle-filled polymer suspensions is an important area of research, since it affects not only the material’s processability but also its final application [1,2]. Depending on filler type and volume fraction, nanoparticle-filled polymer suspensions form a percolating polymer-filler network that may be broken-up through shear stress [3]. Consequently, the rheological behavior of such suspensions is largely influenced by filler-filler and polymer-filler interactions. In particular, an additive manufacturing (AM) technology, known as direct-ink-writing (DIW), uses particle-filled polymer suspensions to fabricate unique, flexible three-dimensional printed objects [4]. The ability of a DIW resin, or “ink,” to adequately print objects depends greatly on its rheological properties. Thus, rheology provides valuable insights when formulating inks for DIW technology, since it informs how well a material withstands shear forces before it begins to flow [5]. More specifically, the ink’s printability depends on the equilibrium storage modulus (G’_eq_), which yields information about the stress at which the material behaves like a viscoelastic solid, and the yield stress (σ_y_), which yields information about the stress at which the material begins to flow. These features ensure not only that the ink can easily flow out of the printing nozzle on demand, but also that it will retain its unique shape while on the build plate of the 3D printer. In summary, a DIW ink needs to behave like a shear-thinning fluid; its yield stress must be exceeded during the printing process, but the ink needs to retain its shape after deposition. There are many different DIW techniques, such as using aqueous-based suspensions and materials that do not require curing to consolidate shape after deposition. For techniques using polymeric materials that require curing (such as those used in this work), processes may involve reactions through UV exposure for inks containing photo-initiators or heat for inks containing specific catalysts [6,7,8].

Polydimethylsiloxane (PDMS) is typically used in the formulation of DIW inks since it exhibits many attractive properties such as low glass transition temperature, high thermal and oxidative stability, high gas permeability, and the ability to act as a blank canvas for functional fillers. For instance, previous studies have used PDMS to formulate DIW inks to print objects able to shield ionizing radiation or to act as environmental sensors [9,10,11]. DIW inks based on PDMS tend to include an active filler in order to exhibit viscoelastic behavior; otherwise, unfilled PDMS acts like a fluid. Fumed silica, which consists of aggregates of amorphous silica nanoparticles, is a favored reinforcing agent [12,13,14,15]. In particular, hydrophilic fumed silica has a high surface area and a large number of silanol groups (free, vicinal, and germinal silanols) that hydrogen-bond to oxygen atoms located in the polysiloxane backbone. Treatment of fumed silica through silylation with alkylchlorosilanes or alkylsilazanes results in replacing silanol groups with hydrophobic ones, thus imparting a hydrophobic character to the filler. Hydrophobic and hydrophilic fumed silica are routinely used as thickening agents in many types of media such as paints, inks, adhesives, and sealants. Numerous rheological studies of PDMS suspensions containing both types of fumed silica have been performed over the years to better understand their reinforcement mechanism [13,16,17]. It has been demonstrated that rheological properties of these mixtures are highly dependent on PDMS chemistry and molecular weight as well as surface chemistry of the silica particles [13].

In the present work, both types of fumed silica were incorporated into PDMS suspensions specifically engineered to be used as inks for DIW technology. Relationships between rheological behavior and storage conditions, or aging environments, were explored in the context of 3D printability. “Aging” herein refers to changes to rheological properties outside of polymer degradation caused by typical chemical degradation such as oxidation and hydrolysis. The present work complements previous studies by this group where the printability of DIW inks was evaluated in terms of chemical composition, lattice-printing structure, and printing parameter [11,18]. It was found that printing parameters such as the strut diameter (or nozzle size) and the spacing ratio of the printed lattice had the most influence on their printability. This study focuses on aging effects on the rheological properties of DIW inks stored for up to 30 days. Suspensions containing hydrophilic and hydrophobic fumed silica at concentrations ranging from 8 wt.% to 25 wt.% were stored under ambient (25 °C and 10% RH), refrigerated (4 °C and 8% RH), and humid (25 °C and 50% RH) conditions. The printability of stored suspensions was evaluated as a function of storage time through rheological experiments, observational data from the DIW printing process, and mechanical testing of the 3D printed samples.

## 2. Materials and Methods

### 2.1. Materials

The suspensions formulated in this study consisted of siloxanes and fumed silica mixed using similar methods previously reported by this team [9,11]. Siloxanes included a vinyl-terminated (4–6% diphenylsiloxane)-dimethylsiloxane copolymer (Gelest PDV-541) and a trimethylsiloxy-terminated methylhydrosiloxane-dimethylsiloxane copolymer (Gelest HMS-301; Gelest, Inc., Morrisville, PA, USA). A high-temperature platinum catalyst (Gelest SIP 6829.2; platinum carbonyl cyclovinylmethylsiloxane complex; 1.85–2.1% Pt in cyclomethyl vinyl siloxanes) was used as the curing agent. To prevent premature curing of the suspensions, 1-ethynyl-1-cyclohexanol (ETCH; 99%, Sigma Aldrich, Millipore Sigma, St. Louis, MO, USA) was utilized. Fillers included OH-functionalized fumed silica (A300; Evonik Aerosil 300; Evonik Industries AG, Essen, Germany; Figure 1a), as well as PDMS-functionalized fumed silica (TS-720; CAB-O-SIL TS-720; Cabot Corporation, Boston, MA, USA; Figure 1b), which were dried under vacuum overnight prior to using them in the suspensions. Specific surface areas (determined via BET analyses) reported by the vendors are 300 m^2^/g for A300 and 120 m^2^/g for TS-720, and average aggregate sizes are 100 nm and 200–300 nm, respectively.

### 2.2. Formulation of Suspensions and Printing Parameters

The basic formulation of all suspensions consisted of a 9:1 ratio of PDV-541 to HMS-301 and <1 wt.% ETCH. Hydrophilic A300 was added in concentrations of 8, 10, and 12 wt.%, while the hydrophobic TS-720 was added in higher concentrations of 14, 16, 18, and 25 wt.% in an attempt to achieve similar rheological properties as suspensions with hydrophilic fumed silica. When TS-720 suspensions were made with the same fumed silica concentrations as A300 suspensions, the resulting resins were too fluid to analyze on the rheometer. For this reason, TS-720 suspensions were all made with higher fumed silica concentrations. To formulate a suspension, HMS-301 and ETCH were initially combined in a disposable plastic cup and mixed with a THINKY planetary vacuum mixer (ARV-310, THINKY USA Inc., Verdugo, CA, USA) under ambient pressure for five minutes at 2000 rpm. After this initial mixing, the designated silica filler was added in the aforementioned weight percentages before adding PDV-541. The combination was then mixed at ambient pressure for another five minutes at 2000 rpm. Prior to DIW printing, the high-temperature platinum catalyst was added to the suspensions to promote hydrosilylation reactions between Si-H groups and vinyl moieties (-CH=CH_2_) [19,20,21] to cross-link the cured printed sample. The inks formulated in this work were cured using an Ossko catalyst, a high-temperature platinum-cyclovinylmethyl-siloxane complex.

For printing, the catalyzed suspensions were first loaded into a 25 mL stainless steel syringe and centrifuged at 2000 rpm for 1 min to remove any air bubbles. The filled syringes were attached to an EMO-XT print head (Hyrel 3D, Atlanta, GA, USA) and then connected to a Hydra 21 3D printer (Hyrel 3D). Inks were extruded onto the build plate at room temperature using 254 µm and 410 µm plastic luer-lock nozzles (Nordson EFD Precision Tips; Nordson Corporation, Westlake, OH, USA). The printer was controlled using Repetrel software (v. 4.2.505; Hyrel 3D) and custom G-code, which specified the face-centered tetragonal (FCT) lattice geometry and center-to-center spacing between struts. Successful printed samples had center-to-center spacing between struts of 500 µm, a travel rate of 2250 mm/min, and a flow rate of 80 pulses/µL. Samples printed with a 254 µm nozzle had layer heights of 225 µm, and samples printed with a 410 µm nozzle had a layer height of 300 µm. These settings were used to ensure adequate adhesion of the first layer to the print bed, consistent extrusion of ink, and slight overlap between consecutive layers for each printed sample. After printing, the samples were cured for 2 h inside a preheated oven at 150 °C.

### 2.3. Characterization

Rheological properties of the suspensions were determined using a TA Discovery Series Hybrid Rheometer DHR-3 (TA Instruments, New Castle, DE, USA). Representative samples of each suspension were tested by the same operator at regular intervals during the 30-day aging process using a 25 mm cross-hatched parallel plate fixture geometry with strain sweeps conducted from 0.001% to 10% strain at an angular frequency of 10 rad/s and stress sweeps conducted from 10 to 10,000 Pa (or until the yield stress was reached) at an angular frequency of 10 rad/s. The temperature of the parallel plate fixtures was set to 25 °C, and the experimental distance of the plates was 1 mm. The yield stress (σ_y_) and equilibrium storage modulus (G’_eq_) of each sample were determined by TA Instruments’ Trios software (v. 5.4.0.300).

A confocal microscope (Keyence VHX-6000; Keyence Corporation, Osaka, Japan) was used to obtain optical images.

An INSTRON 3343 Low-Force Testing System (INSTRON; Norwood, MA, USA) with BlueHill Universal software (v. 4.08) was used to perform uniaxial compression testing on all printed samples. Each sample was subjected to 4 cycles of compression to a maximum stress of 0.6 MPa at a rate of 0.05 mm/s. The stress-strain curve for each printed sample was determined by the final cycle, and the displacement reported was corrected based on when the instrument detected force applied due to contact with the sample.

Dynamic mechanical analysis (DMA) was conducted to observe storage modulus (G’), loss modulus (G”), and loss tangent (tan δ) values of printed samples using a frequency sweep experiment. Cylindrical 8 mm diameter pucks were punched out of full samples and were subjected to a frequency sweep between 0.05 Hz and 200 Hz at a strain rate of 0.025% with a preload value of 0.15 N. All experiments used a set environmental temperature of 25 °C with 10 data points collected in each decade.

### 2.4. Storage Environments

Rheological experiments were performed immediately after the suspensions were prepared. These pristine suspensions, named “control” herein, were subsequently placed in polypropylene THINKY cups and exposed to their unique storage environments for 30 days. The storage environments included: an ambient lab environment (in a high desert region in Los Alamos, NM, USA) with an average temperature of 25 °C and an average RH of 10%, a refrigerated environment with an average temperature of 4 °C and an average RH of 8%, and a humid environment (humidity chamber) with an average temperature of 25 °C and an average RH of 50%. Temperature and humidity levels were measured with a Fisherbrand Traceable Thermometer (Fisher Scientfic, Hampton, NH, USA). Rheological properties of the suspensions stored in these environments were evaluated throughout the 30-day storage period.

## 3. Results and Discussion

### 3.1. Rheolological Properties of Suspensions Containing Hydrophilic and Hydrophobic Fumed Silica

Rheological experiments were performed to evaluate the response of control suspensions to applied shear stress. Irrespective of the surface characteristics of silica, silica-filled PDMS suspensions undergo a liquid-gel transition at some critical filler concentration [17]. Essentially, at low filler concentrations, storage modulus (G’) remains lower than loss modulus (G”), and no plateau region is observed. As the filler concentration increases, there is a crossover of G’ and G”; G’ becomes larger than G”, and a linear viscoelastic behavior characterized by a plateau region in both moduli may appear. In this region, G’ and G” have constant values, which are labeled herein as G’_eq_ and G”_eq_, respectively. Such rheological properties—plateau region and G’ > G”—are features observed for all suspensions formulated in this work, either containing hydrophobic or hydrophilic fumed silica (Figure 2). This rheological behavior indicates a gel-like nature, a condition necessary for 3D printing, since such suspensions are more likely to hold their shape after being extruded from the printing nozzle [22]. Increasing filler concentration leads to higher G’_eq_ and G”_eq_ values for both types of suspensions, demonstrating the reinforcing effect of fumed silica, which disperses well in PDMS following physical mixing [12,16,23]. The reduction in surface silanols on treated silica, such as TS-720, also reduces particle-particle and particle-filler interaction. Thus, the reinforcing effect is much more pronounced for A300-containing suspensions than for TS-720 suspensions due to A300’s higher specific surface area and the presence of hydroxyl groups on its surface leading to hydrogen bonds with the polymer chain. For instance, G’_eq_ is 194 kPa for the suspension containing 12 wt.% A300 but only 59 kPa for the suspension containing 18 wt.% TS-720. Ultimately, lower G’_eq_ values equate to easier deformation of the polymer-filler network formed in the suspension. Consequently, suspensions formulated with hydrophobic fumed silica are more susceptible to deformation at lower applied stresses than those formulated with hydrophilic fumed silica.

As oscillation stress increases, a steep decrease in G’ coinciding with a G” maximum is observed for A300 suspensions (Figure 2a). This effect, known as the Payne effect, was first observed for carbon black filled rubbers, and it is largely reduced for less active fillers [24,25]. Consequently, it is less notable for the TS-720 suspensions (Figure 2b). The oscillation stress at which G’ and G” meet defines the gel point of the suspension, and, as shown in Figure 2, it depends on silica concentration and surface chemistry. The stress at which G’ begins a steep decline characterizes the yield stress. It occurs at 6 kPa for the 12 wt.% A300 suspension and 0.20 kPa for the 18 wt.% TS-720 suspension. The yield stress of a suspension is an important parameter for DIW, since it indicates the minimum stress needed to observe deformation of the polymer network, or in other words, the minimum pressure required to cause the suspension to start flowing out of the printing nozzle.

### 3.2. Effects of Storage Environment on Suspensions Containing A300 Fumed Silica

Figure 3 shows G’ and G” curves versus oscillation stress obtained for stored suspensions containing 8, 10, and 12 wt.% A300 fumed silica. Suspensions were tested after being stored for 30 days either in a refrigerator (3 °C and 8% RH), on a lab benchtop (25 °C and 8% RH), or in a humidity chamber (25 °C and 50% RH). Compared to control suspensions, aged samples showed a decrease in G’_eq_ and an increase in G”_eq_ (see Table 1). This result is more pronounced for suspensions containing high filler concentrations. For instance, the 8 wt.% A300 suspension stored for 30 days under ambient conditions showed a decrease in G’_eq_ of about 29%, whereas the 12 wt.% A300 suspension showed a decrease in G’_eq_ of 54%. These trends indicate a shift from a gel-like nature to a more fluid-like character. This observation is noteworthy, since it represents the opposite of creep hardening, which is a phenomenon commonly observed in silica-filled siloxane mixtures. In the case of the A300 suspensions in this study, polymer chains tend to wet the silica surface as time progresses, increasing the number of bridging chains between filler particles. These bridging chains work to drag silica particles together, which may cause agglomeration and the subsequent removal of reinforcement from much of the suspension leading to a decrease in storage modulus. This phenomenon is typically observed for suspensions with low viscosity, which allow filler particles to move easily through the suspension. Creep hardening, on the other hand, takes place in suspensions with very high viscosity; in this case, filler agglomeration is hindered leading to aged suspensions with higher storage moduli than unaged ones [26].

Furthermore, the settling effects of silica particles during the storage of the formulation was investigated. Rheological experiments were performed on samples collected from the top and bottom of the storage container. These results, shown in Figure 3d, confirmed that the changes in the rheological properties of the mixed samples are not due to settling effects but are due to the agglomeration of the fillers, as discussed above.

It is also informative to examine changes to rheological properties in terms of tan δ, which represents the ratio between G” and G’. Tan δ values are listed in Table 1 for the plateau region. Aged suspensions exhibit higher tan δ values than control suspensions, indicating once again a higher degree of filler aggregation with aging [16]. When filler dispersion in the polymer matrix is reduced, there are fewer restrictions against molecular motion of polymer chains, resulting in a less elastic response of the material, as evidenced by higher tan δ values. This increase in tan δ is more pronounced for suspensions stored in the humidity chamber, followed by ambient conditions and then suspensions stored under refrigerated conditions. In summary, rheological properties of suspensions containing hydrophilic fumed silica are heavily influenced by environmental conditions with the high humidity storage environment being the worst condition investigated here. This is likely due to water adsorption on the surface of fumed silica working to increase hydrogen bonding between silica particles, thus decreasing silica dispersion in the siloxane matrix.

### 3.3. Effects of Storage Environment on Suspensions Containing TS-720 Fumed Silica

Figure 4 shows G’ and G” curves versus oscillation stress for stored suspensions containing 14, 16, 18, and 25 wt.% TS-720 fumed silica. Although these suspensions show similar rheological behavior as previously discussed for A300 suspensions (a plateau region and G’ > G” at low oscillation stresses), moduli values are much lower than those for A300 suspensions. The only exception is the suspension containing the highest TS-720 concentration, or 25 wt.%, which exhibits similar rheological behavior to the 12 wt.% A300 suspension. These results are consistent with the observation that a higher percolation concentration is required for the formation of filler networks within PDMS as the silica surface becomes more inactive (TS-720 surface is treated with PDMS, making it hydrophobic) [16]. As it was found for A300 suspensions, TS-720 suspensions also display changes to rheological properties with storage conditions, as shown in Table 2. The largest changes were observed for the suspensions stored in the humid environment; tan δ for the suspension containing 14 wt.% TS-720 almost doubled in value. These results are interesting considering the fact that the full surface treatment on the silica surface should have heavily reduced polymer adsorption and the consequent formation of bridging chains. It is interesting to compare these results with DeGroot and Macosko’s work on the effects of aging on the rheological properties of PDMS suspensions containing hydrophobic silica [26]. These authors reported that the amount of PDMS bound to the silica surface was independent of the aging time, and the *G’_eq_* value remained stable over a two-month period irrespective of the molecular weight of PDMS. This indicated that polymer adsorption reached equilibrium very quickly, and few bridging chains were formed by the full surface treatment of fumed silica. Differing results in this study may be due to the distinctive mixing methods (planetary mixer versus recirculating screw mixer) and the dehydration versus a lack of pretreatment of silica before use. Thus, we infer that initial trapped water residues in the silica aggregates used here and the differences in mixing procedures contributed to the observed changes in G’ and tan δ values over time.

### 3.4. Printability of Control Suspensions

Printability is the ability of a suspension to retain its shape during the DIW process until it is ready to be cured. As shown in Figure 5, the most successful printed samples using freshly formulated 10 and 12 wt.% A300 suspensions produced well-defined strut patterns and highly detailed prints, which confirm the effectiveness of A300 suspensions to act as a DIW ink. On the other hand, the suspension containing only 8 wt.% A300 was not able to effectively print the FCT structure as there was a great deal of poor bridging areas (strut sagging over gaps) and poor print quality overall. Suspensions containing 14, 16, and 18 wt.% TS-720 produced prints with no strut definition and overall poor-quality printed samples, as shown in Figure 6. The only suspension containing TS-720 that was able to produce acceptable prints was the one containing 25 wt.% TS-720. However, due to nozzle clogging, this suspension had to be printed using a 410 µm nozzle instead of the 254 µm nozzle used for the A300 and lower concentration TS-720 suspensions. Because the 8 wt.% A300 and 14, 16, and 18 wt.% TS-720 suspensions did not produce good quality prints using the control inks, the printability of their aged inks was not investigated.

### 3.5. Printability of Stored Suspensions

A300 suspensions stored for 30 days under ambient, refrigerated, and humid conditions clearly showed degradation of their overall printability (Figure 7 and Figure 8). Stored samples showed a drastic decrease in how well the suspension was able to stay in the defined strut patterns once extruded from the printing nozzle. The individual layers of the printed aged suspensions began to flow together to form a singular unit rather than a three-dimensional lattice structure, which indicates a significant decrease in printability. Comparatively, layers were clearly distinguishable in the prints made with the same freshly prepared control suspensions. Figure 7 shows samples printed using the 10 wt.% A300 silica suspension that was stored under the three investigated environmental conditions. The cross-sectional images provide a visualization of the reduction in printability due to aging represented by undefined layers and combined struts compared to what was observed for the control suspension (Figure 5). This reduction in printability can be attributed to the more fluid-like nature of the aged suspensions indicated by the rheological analyses. One intriguing result is the increased transparency in the printed sample made from the resin that was stored in the humid environment. The increased transparency could be attributed to filler agglomeration [12,27]. This print also appears to have a large number of bubbles that seem to be a direct result of water evaporation. Overall, the hydrophilic nature of A300 fumed silica and the high water vapor permeability of PDMS suggest that the moisture in the humid environment was the key contributor to the decreased printability of the 10 wt.% A300 suspensions [28]. Consequently, if a hydrophilic silica-based polymer suspension must be stored for an extended period, a humid environment should be avoided.

While the overall printability was greatly reduced for 10 wt.% A300 suspensions after storage, one viable option for stability during longer storage is increasing the silica concentration. Figure 8 shows 12 wt.% A300 silica samples, and one printed sample stands out: the printed sample that was stored under ambient conditions. This printed sample has somewhat identifiable struts and distinguishable layers. Alongside this print, the suspensions stored under refrigerated and humid conditions have many fewer identifiable struts along with poor bridging and cannot be categorized as good quality prints. Therefore, along with increasing the fumed silica concentration, storing at roughly room temperature and under lower humidity environments appear to be necessary for increasing the shelf-life of hydrophilic silica DIW suspensions.

As for suspensions containing TS-720 fumed silica, only the 25 wt.% ink was printable; when stored under these three environmental conditions, the printability remained relatively unchanged, as shown in Figure 9. The most plausible reason for this retention of printability was the hydrophobic nature of the TS-720 fumed silica. The hydrophobic fumed silica reduced environmental interactions (i.e., moisture uptake) with the suspension compared to the hydrophilic suspensions, which allowed for the retention of its DIW-relevant rheological properties and printability.

It is worth mentioning that the polymer-filler network may be negatively disturbed once the suspension is extruded from the printing nozzle, and such extruded suspension may have different rheological properties that may change its stability on the build plate. Thus, rheological properties were investigated for 8 and 12 wt.% A300 suspensions after they were extruded through the printing nozzle. Figure 10 shows that the extruded suspensions maintain their initial rheological properties indicating that extrusion from the nozzle during printing does not have a significant effect on the rheology of the suspension.

Next, the trends in changes to rheological properties over time are examined. Figure 11 shows tan δ versus time for printable suspensions (10 and 12 wt.% A300 and 25 wt.% TS-720). It is clear that tan δ values changed the least when suspensions were stored in a refrigerated environment. In addition, assuming the cut-off for ideal printability to be the tan δ observed for the control 8 wt.% A300 suspension (i.e., tan δ being approximately equal to 0.05), then 10 and 12 wt.% A300 suspensions are printable only when stored for periods of time shorter than 30 days (Figure 11a,b). As for the 25 wt.% TS-720 suspension, the tan δ cut-off value for printability was set to 0.36 (which represents the highest tan δ value observed for the suspension stored for 30 days in a humid environment and remains printable). This indicates that this suspension can be stored for up to 30 days without loss of printability. The control 12 wt.% A300 and 25 wt.% TS-720 suspensions both had G’ values around 200 kPa, but their tan δ values differ by an order of magnitude. This is because the G” of the 25 wt.% TS-720 suspension is over 7 times greater than that of the 12 wt.% A300 suspension, leading to a much larger tan δ value. Additionally, since G” represents the “viscous” component of the viscoelastic suspensions, the rheological results suggest that the 25 wt.% TS-720 suspension is much more viscous than the 12 wt.% A300 suspension. This was confirmed when printing the two suspensions because 12 wt.% A300 could be printed with a 254 µm nozzle while 25 wt.% TS-720 suspension required the use of a 410 µm nozzle to avoid clogging issues.

Another useful approach to verify the printability of aged suspensions is to perform mechanical experiments on the 3D printed pads. Because the rheology and printability of the 25 wt.% TS-720 suspension remained the most stable in all storage environments, the compressive stress-strain responses of samples printed with control and stored 25 wt.% TS-720 suspensions were investigated. As seen in Figure 12, no changes to mechanical properties were observed, once more confirming the robustness of this suspension under multiple storage conditions. The mechanical response of a sample printed with the control 12 wt.% A300 suspension is also provided in the same figure as a comparison. The differences in mechanical response for 12 wt.% A300 and 25 wt.% TS-720 can be explained by the increased filler concentration present in the TS-720 formulation. DMA results of the same printed samples confirm compressive stress-strain behavior.

## 4. Conclusions

This study investigated the effects of storage in varying environmental conditions on the printability of fumed silica filled PDMS suspensions. The novelty of this work lies in the analyses of the interconnected effects of storage environment, fumed silica composition/concentration, rheology, and printability of these suspensions in the context of additive manufacturing. The most interesting rheological finding was the overall increase in tan δ at the viscoelastic plateau region with increasing storage time for all storage conditions, fumed silica compositions, and filler concentrations. These changes in rheological properties of the stored suspensions resulted in a decreased ability to 3D print FCT lattice structures via DIW. The suspension least affected by storage conditions was the one containing the highest concentration of hydrophobic fumed silica. This suspension, containing 25 wt.% TS-720 fumed silica, maintained its printability even after being stored for 30 days in environments with high and low relative humidities and temperatures. The rheological properties of this suspension as well as the lattice structure and mechanical properties of samples printed with it all confirmed its ability to be stored in multiple conditions without losing functionality. Additionally, an increased fumed silica concentration can assist in counteracting any deficits that accompany long-term storage. As a result, in the event a silica-filled polymer suspension to be employed in DIW technology must be stored for an extended period of time, the best method to enhance its shelf life is to use hydrophobic fumed silica fillers, increase the fumed silica (hydrophobic or hydrophilic) concentration, and/or store the suspension in a low-humidity, low-temperature environment.

## Figures and Tables

**Figure 1 polymers-15-04334-f001:**
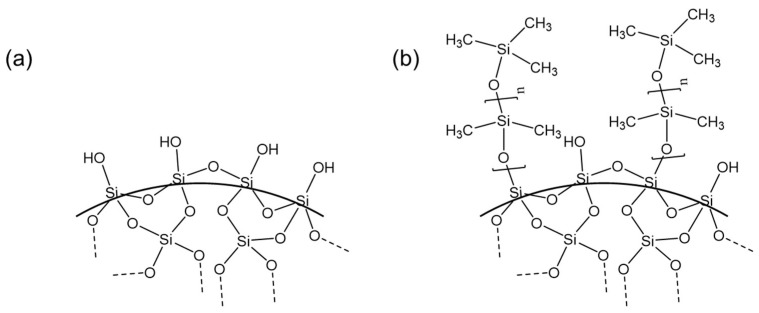
Schematic diagram for the structure of fumed silica: (**a**) A300 (hydrophilic fumed silica) and (**b**) TS-720 (PDMS treated hydrophobic fumed silica).

**Figure 2 polymers-15-04334-f002:**
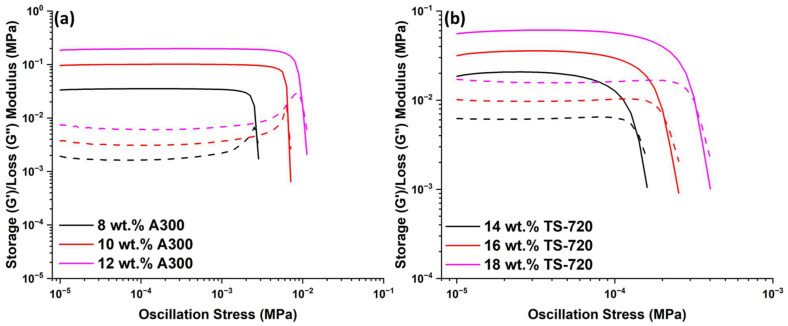
G’ (solid-line) and G” (dashed-line) of suspensions containing (**a**) 8, 10, and 12 wt.% A300 and (**b**) 14, 16, and 18 wt.% TS-720.

**Figure 3 polymers-15-04334-f003:**
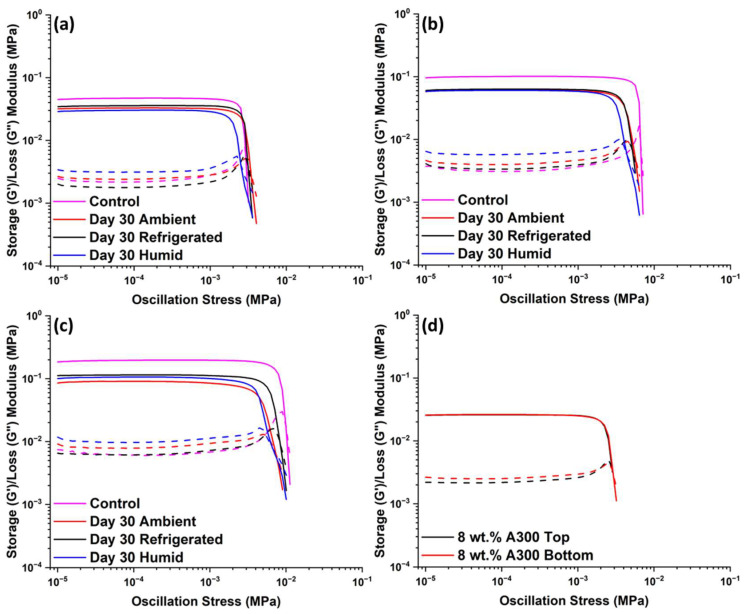
G’ (solid-line) and G” (dashed-line) of suspensions stored for 30 days in ambient, refrigerated, or humid conditions; (**a**) 8 wt.% A300 fumed silica, (**b**) 10 wt.% A300 fumed silica, (**c**) 12 wt.% of A300 fumed silica, and (**d**) 8 wt.% A300 suspensions based on the location in which the suspension was taken from the storage container.

**Figure 4 polymers-15-04334-f004:**
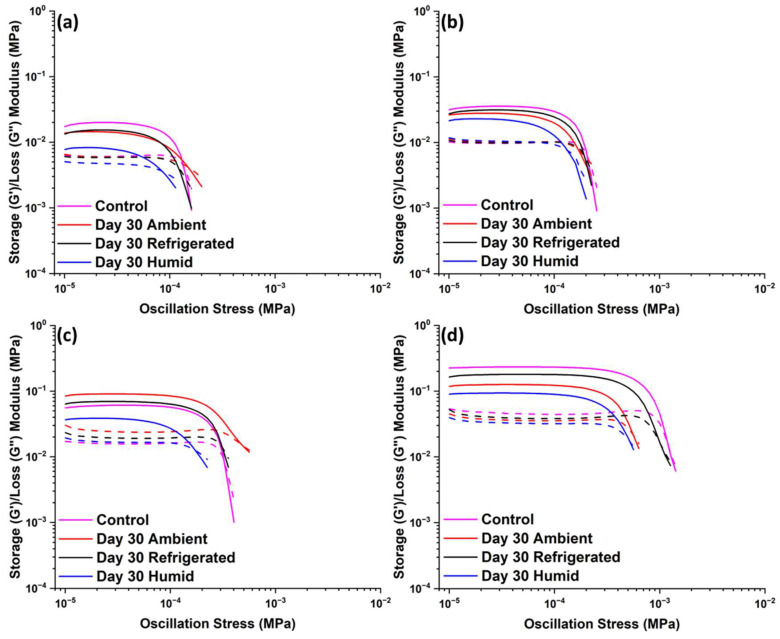
G’ (solid-line) and G” (dashed-line) of suspensions stored for 30 days in ambient, refrigerated, or humid conditions; (**a**) 14 wt.% of TS-720 fumed silica, (**b**) 16 wt.% of TS-720 fumed silica, (**c**) 18 wt.% of TS-720 fumed silica, and (**d**) 25 wt.% of TS-720 fumed silica.

**Figure 5 polymers-15-04334-f005:**
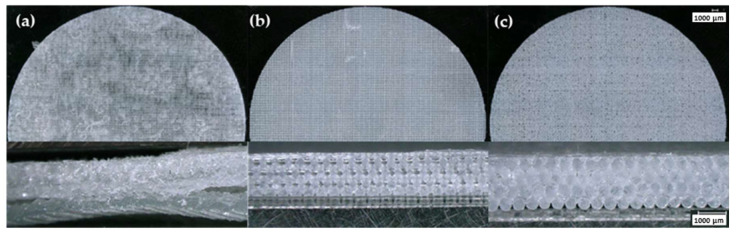
Optical microscopy images showing top view and cross-section of samples printed with (**a**) 8 wt.%, (**b**) 10 wt.%, and (**c**) 12 wt.% A300 control suspensions.

**Figure 6 polymers-15-04334-f006:**
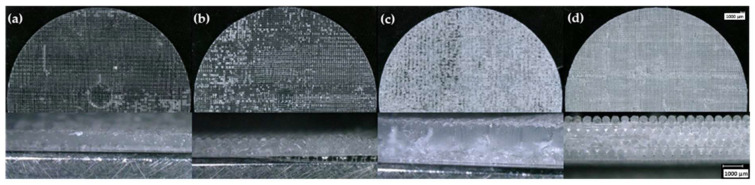
Optical microscopy images showing top view and cross-section of samples printed with (**a**) 14 wt.%, (**b**) 16 wt.%, (**c**) 18 wt.%, and (**d**) 25 wt.% TS-720 control suspensions.

**Figure 7 polymers-15-04334-f007:**
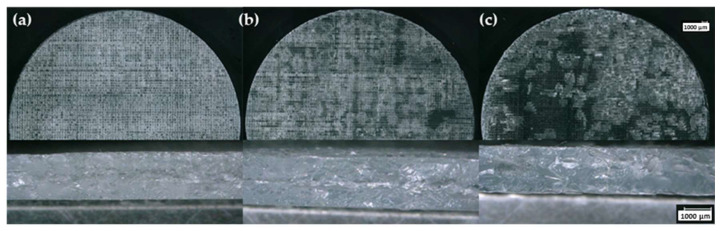
Optical microscopy images of 10 wt.% A300 silica samples formulated with suspensions stored at (**a**) 25 °C and 10% RH, (**b**) 4 °C and 8% RH, and (**c**) 25 °C and 50% RH for 30 days.

**Figure 8 polymers-15-04334-f008:**
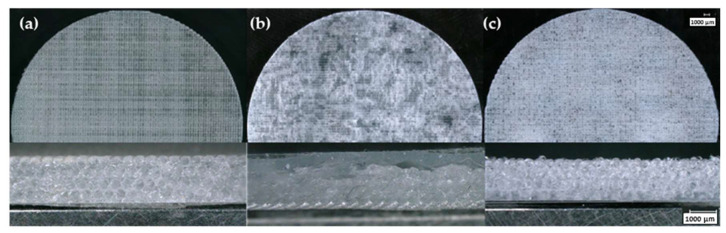
Optical microscopy images of 12 wt.% A300 silica samples formulated with suspensions stored at (**a**) 25 °C and 10% RH, (**b**) 4 °C and 8% RH, and (**c**) 25 °C and 50% RH for 30 days.

**Figure 9 polymers-15-04334-f009:**
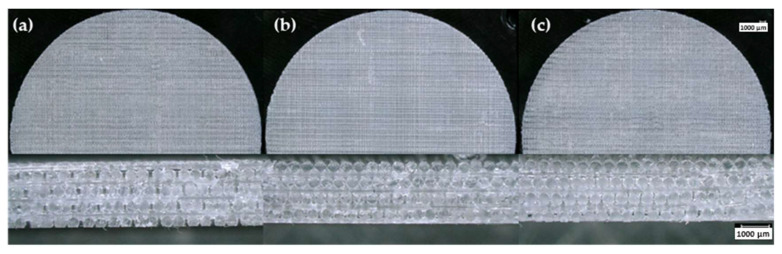
Optical microscopy images of 25 wt.% TS-720 silica samples formulated with suspensions stored at (**a**) 25 °C and 10% RH, (**b**) 4 °C and 8% RH, and (**c**) 25 °C and 50% RH for 30 days.

**Figure 10 polymers-15-04334-f010:**
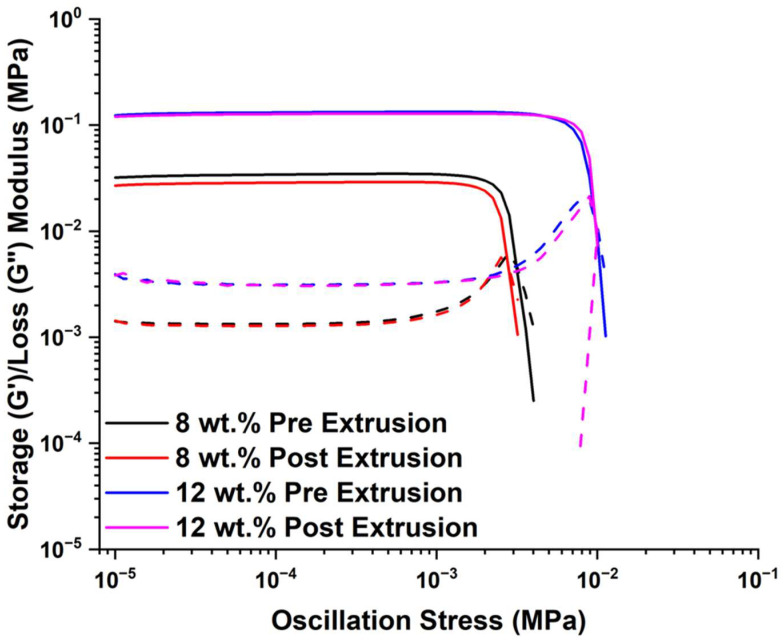
G’ (solid-line) and G” (dashed-line) of A300 suspensions before and after being extruded by the 254 µm priting nozzle.

**Figure 11 polymers-15-04334-f011:**
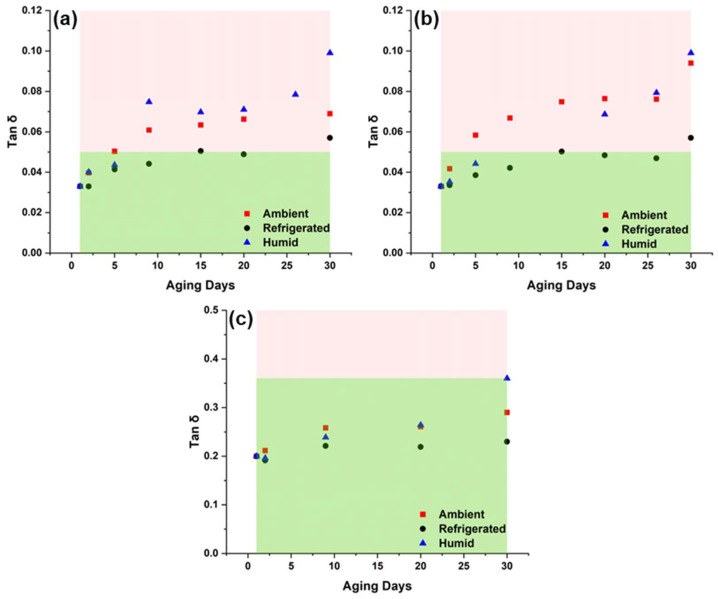
Tan δ versus storage time for (**a**) 10 wt.% A300, (**b**) 12 wt.% A300, and (**c**) 25 wt.% TS-720 suspensions stored in ambient, refrigerated, and humid environmental conditions. Red shaded areas represent poor printability, and green shaded areas represent good printability. The cut-off lines in (**a**) and (**b**) represent tan δ of the control 8 wt.% A300 suspension, which was not printable. The cut-off line in (**c**) represents the highest tan δ value observed for a 25 wt.% TS-720 suspension stored for 30 days and remained printable.

**Figure 12 polymers-15-04334-f012:**
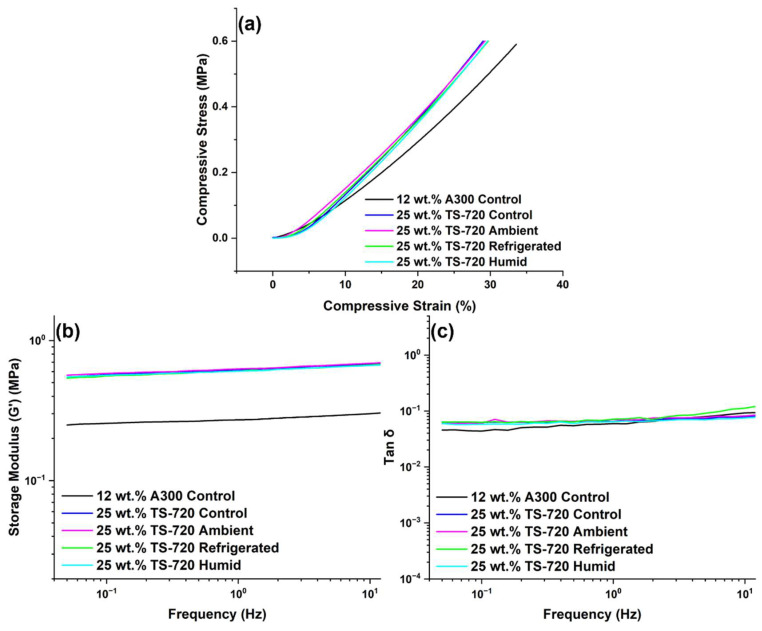
Mechanical response of 3D printed pads using 25 wt.% TS-720 and 12 wt.% A300 suspensions: (**a**) compressive stress-strain curve, (**b**) storage modulus, and (**c**) tan δ obtained by DMA experiments.

**Table 1 polymers-15-04334-t001:** G’_eq_ (kPa), G”_eq_ (kPa), and tan δ (G”_eq_/G’_eq_) values of control and 30-day aged suspensions containing varying wt.% of A300 fumed silica.

A300	Control			Ambient			Refrigerated			Humid		
	G’_eq_	G”_eq_	tan δ	G’_eq_	G”_eq_	tan δ	G’_eq_	G”_eq_	tan δ	G’_eq_	G”_eq_	tan δ
8 wt.%	46.3	2.3	0.050	32.6	2.5	0.078	35.6	1.84	0.05	29.9	3.2	0.107
10 wt.%	99.9	3.3	0.033	59.9	4.1	0.069	62.5	3.57	0.06	59.7	5.9	0.099
12 wt.%	194.4	6.5	0.033	88.0	8.3	0.094	113.6	6.59	0.06	104.0	10.3	0.099

**Table 2 polymers-15-04334-t002:** G’_eq_ (kPa), G”_eq_ (kPa), and tan δ (G”_eq_/G’_eq_) values of 30-day aged suspensions containing varying wt.% of TS-720 fumed silica.

TS-720	Control			Ambient			Refrigerated			Humid		
	G’_eq_	G”_eq_	tan δ	G’_eq_	G”_eq_	tan δ	G’_eq_	G”_eq_	tan δ	G’_eq_	G”_eq_	tan δ
14 wt.%	19.0	6.1	0.32	13.8	6.00	0.43	14.5	5.9	0.41	7.77	4.7	0.61
16 wt.%	33.8	9.9	0.29	27.1	10.4	0.38	29.9	10.0	0.33	21.9	10.6	0.48
18 wt.%	59.0	16.1	0.27	88.3	24.8	0.28	66.7	19.8	0.30	36.9	17.1	0.46
25 wt.%	227.0	46.4	0.20	125.0	36.7	0.29	177.0	39.9	0.23	92.1	33.6	0.36

## Data Availability

The authors confirm that the data supporting the findings of this study are available within the article.

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
