# Peer review of "Interplay between Shelf Life and Printability of Silica-Filled Suspensions"

_polymers, 2023, doi:10.3390/polym15214334_

Round 1
Reviewer 1 Report
Comments and Suggestions for Authors
Reviewer 2 Report
Comments and Suggestions for Authors
Introduction
The introduction should be slightly improved. the authors presented their "printing way" as if it was the only way to print using the DIW method. But DIW also uses aqueous-based suspensions and does not always require curing to consolidate shape. So, the authors should write that their way is one among others in DIW method.
Line 50, “…… we have used..”- it would be more appropriate to use the impersonal form
Materials
Not only specific surface area matters but also size and particle size distributions influence the rheological properties. The authors should provide such information about used fillers.
Storage environments
Please specify the material of containers in which the pastes were sealed. Different polymers have different oxygen and water (vapor) permeability coefficients which might influence storage. Comment this in the text. Give information about the sealing procedure e.g. application of parafilm etc.
Line 159. The ambient RH lies between 40-55%. In this manuscript, ambient RH is 10%. Why this value is so low?
Results and discussion
Line 183 and in the rest of the manuscript; “……demonstrating the reinforcing effect of fumed silica.” - by definition, viscosity is the internal friction of a fluid, thus when you increase the volume fraction of powder you increase the internal friction (particle-particle interaction, etc), and in such a way rheological properties of system. Thus it is hard to say that something was reinforced. It would be better to use another expression.
Line 183-190; This part should be improved. Rheological properties are influenced by the chemistry of polymers and the surface of silica powders but mostly, by the powder's “conditions”: morphology, size distribution, and agglomeration. The authors should explain that based on chemistry and interaction between PDMS and silicas surface: presence of OH group or PDMS and possibility to form e.g., hydrogen bonds, etc. What is more, they should refer here to the size distribution of powders and to the SEM images of such powders (agglomeration of powders might be visible). The best if the authors could provide information about the homogeneity of their materials e.g. SEM of silica in cured polymeric matrix. lease,
Line 218 , Does “lab condition” means ambient (Table 1)? Please, keep uniform nomenclature
Line 221-230 . What is the author's explanation for this phenomenon?
Line 234 and Figure 2. There is no Figure 2d!!!
Line 254 ; Do the authors measure the H2O concentration or investigate the presence of the extent of water after storage in a humid environment? Do the authors have any proof of water adsorption? Please, provide such proof
Line 275; I do not agree. The percolation is related to the size distribution and/ or agglomeration and/or homogenization of suspension. The PDMS functionalization affects the rheological properties but it can not be written that “high percolation concentration is required for the formation of filler network as the silica surface becomes more inactive”
Line 353; the authors wrote, “The hydrophobic fumed silica reduced environmental interactions with the suspension, which allowed for the retention of its printability.”. Please, be more specific here. Do the authors mean that e.g. tan delta is decreased less in the case of suspensions containing TS-720? What is the more important parameter that allowed us to relate rheological properties with printability?
Reviewer 3 Report
Comments and Suggestions for Authors
Reviewers comment:
1. Authors report the well studied systems. Systems reported are not new. Rheological analysis presented is not new. The suspensions based on different silicas i.e. OH functionalised fumed silica (A300) and PDMS functionalise fumes silica (TS720) are new and not reported before. However, in the manuscript authors do not discuss fundamental differences and impact of OH vs. PDMA functionalisation.
In addition to data presentation more scientific discussion would be helpful for the readers of POLYMERS.
2. It is recommend to explain the interaction and difference of A300 and TS720 via. Schematic as figure1.
3. In the manuscript, A300 was added in concentrations of 8, 10, and 12 wt.% while the 102 TS-720 was added in higher concentrations of 14, 16, 18, and 25 wt.%. It is highly recommended to report both systems using at least on same/ common conc. and report rheological analysis of such system.
4. Can authors also comment on how the rheological system would behave when A300 and TS720 are mixed in suspensions e.g. A300/TS720: 50%/50%.
5. Such system would enable to demonstrate is there any preferential interaction with OH vs. PDMS functionalisation.
6. In addition to rheological analysis preferential interaction can also be analysed using FTIR or other spectroscopic technique, which are direct indication of interaction.
7. Fundamental discussion and novelty should be detailed in conclusion.
Some other minor corrections needed for reference
· Figure 10. Formatting of this is figure is not consistent with formatting of other figures in the manuscript. Authors are requested to revise the formatting of figure 10 to match with formatting of other figures in the manuscript.
· On page 15 and line 246
..filler aggregation with aging.16 When the degree.. should be corrected as.. filler aggregation with aging16. When the degree..
· on page 7 and line 282
with DeGroot and Macosko’s work 25 on the reference should be mentioned at the end of sentence as pe the format followed in other part of manuscript.
· legends front size in figure 1 a and b are not the same.

Round 2
Reviewer 1 Report
Comments and Suggestions for Authors
I thank the authors for addressing the comments that they could but would have appreciated more clarity about my comments that they did not.
I especially appreciate Figure 11.
A comment. In general, if the response is that "multiple prints", "multiple sample" etc were tested, but a number is not provided, why not? was this not tracked? I understand taking averages, thats perfect, but that means that multiple values are available to average, why not provide the SD or SEM? I think the science is sound, Im more concerned about repeatability and methodological robustness. A general lack of statistical treatment is unfortunate, as it could make the claims stronger and help others looking to build on this work.
Author Response
We thank the reviewer for their comments. We carefully looked at our lab notebooks to provide the best possible answer to the reviewer. As previously mentioned, multiple prints of each resin were attempted using various printing parameters before determining the most successful print settings. When a resin was deemed not printable, it meant that at least three tries were attempted before determining that the resin was either not printable or that it produced prints of poor quality. We always strive to provide the maximum possible amount of experimental detail to the reader so that each experiment and sample preparation can be duplicated. Thus, the conclusions of this work reflect that.
Reviewer 2 Report
Comments and Suggestions for Authors
The introduced corrections as well as provided explanations are appropriate.
Please check the numbering in the reference list from No. 5, because the new reference is merged here with the old one, and something might be missing. Please, check the correctness of other literature references.
Apart from that, I have no other comments and the article can be printed in its current form.
Author Response
Thank you for the careful reading of our manuscript. The format of reference 5 was corrected as well as all other references cited in the revised manuscript.
Reviewer 3 Report
Comments and Suggestions for Authors
Authors response accepted.
Author Response
We thank the reviewer for their suggestions and comments.